# Anti-Foulant Ultrafiltration Polymer Composite Membranes Incorporated with Composite Activated Carbon/Chitosan and Activated Carbon/Thiolated Chitosan with Enhanced Hydrophilicity

**DOI:** 10.3390/membranes11110827

**Published:** 2021-10-27

**Authors:** Syeda Samia Nayab, M. Asad Abbas, Shehla Mushtaq, Bilal Khan Niazi, Mehwish Batool, Gul Shehnaz, Naveed Ahmad, Nasir M. Ahmad

**Affiliations:** 1School of Chemical and Materials Engineering, National University of Science and Technology, Islamabad 44000, Pakistan; ssamia.nse5scme@student.nust.edu.pk (S.S.N.); engr.ma.abbas@live.com (M.A.A.); m.b.k.niazi@scme.nust.edu.pk (B.K.N.); 2School of Natural Sciences, National University of Science and Technology, Islamabad 44000, Pakistan; shehla.mushtaq@sns.nust.edu.pk; 3Department of Chemical Engineering, COMSATS University Islamabad, Lahore Campus, Lahore 54000, Pakistan; mehwishbatool@cuilahore.edu.pk; 4Department of Pharmacy, Quaid-e-Azam University, Islamabad 44000, Pakistan; gshahnaz@qau.edu.pk (G.S.); natanoli@qau.edu.pk (N.A.)

**Keywords:** ultrafiltration, polyethersulfone, activated carbon, chitosan, thiolated chitosan, anti-fouling

## Abstract

A rapid increase in population worldwide is giving rise to the severe problem of safe drinking water availability, necessitating the search for solutions that are effective and economical. For this purpose, membrane technology has shown a lot of promise but faces the challenge of fouling, leading to a reduction in its lifetime. In this study, ultrafiltration polyethersulfone membranes were synthesized in two different concentrations, 16% wt. and 20% wt., using the phase inversion method. Chitosan and activated carbon were incorporated as individual fillers and then as composites in both the concentrations. A novel thiolated chitosan/activated carbon composite was introduced into a polyethersulfone membrane matrix. The membranes were then analyzed using Attenuated Total Reflection–Fourier-Transform Infrared spectroscopy(ATR-FTIR), Scanning Electron Microscopy (SEM), optical profilometry, gravimetric analysis, water retention, mechanical testing and contact angle. For membranes with the novel thiolated chitosan/activated carbon composite, Scanning Electron Microscopy micrographs showed better channels, indicating a better permeability possibility, reiterated by the flux rate results. The flux rate and bovine serum albumin flux were also assessed, and the results showed an increase from 105 L/m^2^h to 114 L/m^2^h for water flux and the antifouling determined by bovine serum albumin flux increased from 23 L/m^2^h to 51 L/m^2^h. The increase in values of water uptake from 22.84% to 76.5% and decrease in contact angle from 64.5 to 55.7 showed a significant increase in the hydrophilic character of the membrane.

## 1. Introduction

Water is the basic foundation for life on earth. However, a rapid increase in population worldwide is giving rise to many issues, safe drinking water availability being the most important one. In Pakistan alone, around 21 million people are not able to get water that is safe enough to drink [1]. The usable reservoirs are limited, which necessitates means that can help in salvaging of the used water along with steps to reduce excess water usage so that more water can be saved and recycled. To ensure water availability to the deprived masses, there is a need to find solutions that are effective and economical to be able to meet this challenge. For this purpose, membrane technology has shown a lot of promise. Membranes are mostly used in separation technology that aims towards the separation of one component present in the mixture from the other one by hindering its permeation [2,3,4]. One of the fractionation techniques used in membrane technology is ultrafiltration [5]. Primarily, ultrafiltration is different from other techniques due to its pore size and operating pressure (0.01–0.1µm and 50–120 psi, respectively). Most of the particulates, proteins, colloids, etc., are filtered out using ultrafiltration membranes [6]. Dissolved solids present in the feed solution cannot be filtered out using ultrafiltration but most of the microbial structures can be. Recovery of critical toxins can also be a benefit of this process. Ultrafiltration is also used as a pretreatment process for nanofiltration and reverse osmosis [7].

One of the biggest challenges membranes face is fouling, which reduces their life time [8]. Fouling is the continuous buildup of materials that are deposited on membrane surfaces, clogging the pores and ultimately resulting in membrane failure [9]. Fouling is a prevalent reason for membrane failure. In the process leading up to fouling, the performance of the membrane keeps on declining as the pores are getting either partially or completely blocked. This results not only in a decreased flux rate but also increased operating pressure [10]. The membranes then either need replacement or treatments that help in removal of the foulants, including intense chemical cleaning, resulting in an overall increase in the cost of the treatment process [11]. A lot of research is ongoing to mitigate this problem and a number of materials have been incorporated in membranes to study their effect on fouling [12]. These materials or additives add to the properties of the membrane. Apart from addressing the problem of fouling, other properties can also be enhanced with the incorporation of additives. Some additives help in increasing the hydrophilicity and flux rate, while others might help in manipulation of the structure. In this study, polyvinylpyrrolidone (PVP), chitosan, thiolated chitosan and activated carbon were used as additives, each bringing its own set of properties. PVP and chitosan not only improve a membrane’s hydrophilicity but also increase its anti-fouling properties [13,14]. The key properties of chitosan are that its almost nontoxic, biocompatible and biodegradable [14]. Chitosan acts as a flocculent, resulting in agglomeration of colloids and thus making their extraction easy [15]. Addition of chitosan increases the wettability property of the membrane; hence, enhancing its efficiency by absorbing less protein and reducing the flux loss [16,17].

Modified thiolated chitosan has shown a range of properties that are usable in multiple fields. Apart from its greater swelling properties that make it desirable in biomedical applications, it also has a better ability to bind with metal ions such as Ni2+ and have acted as sorbents for arsenic removal from ground water [18,19,20]. It also adds value to this polymer in that it also has shown inhibition to certain enzymes that can be manipulated in membranes to increase anti-fouling and anti-bacterial activities [18]. The sulfhydryl group added in thiolated chitosan also has a hydrophilic nature, making it more practical for use in membranes. Activated carbon maintains the taste and odor of the water and also acts as excellent adsorbents that also help in reducing fouling [19]. Apart from the presence of enhanced active sites, the presence of van der Waal’s forces also play a role in the physisorption. These forces attract the pollutant from out of the solution and onto the porous surface of the activated carbon [20].

Miao et al. reported polysulfone membranes incorporated with sulfated chitosan having a flux rate of 22.9 L/m^2^h at 0.40 MPa pressure [21]. Ingole et al. prepared polysulfone membranes with activated carbon with a flux rate ranging from 40 to 82 kg/m^2^h for different concentrations of activated carbon [22]. In a study presented by Xiaowei et al., a graphene oxide/chitosan composite was used in mixed matrix membranes and showed a flux rate of 30 L/m^2^h with 3% wt. chitosan and 1 %wt. graphene oxide content [23]. An activated carbon nanoparticles/chitosan composite was incorporated into nanofiltration polyether sulfone membranes by E. Bagheripour et al. and the water flux was measured at different concentrations of the composite. The highest flux was shown by membranes incorporating the 0.5 % wt. composite at 30 L/m^2^h [24].

This study focuses on improvement of the flux rate and anti-fouling properties of polyethersulfone (PES) ultrafiltration membranes by incorporating it with a chitosan/activated carbon composite and novel thiolated chitosan/activated carbon composite. The membranes were prepared under two different PES concentrations (16 and 20 % wt.) and were incorporated in the composites. The fabricated membranes were evaluated using ATR-FTIR and SEM. Gravimetric analysis was done to calculate the mean porosity. The water retention and contact angle were measured for hydrophilicity. The water flux and bovine serum albumin (BSA) flux were also calculated for the flux rate and antifouling properties, respectively. Surface roughness was also examined. A tensile test was performed to evaluate the mechanical strength of the membranes.

## 2. Materials and Methods

### 2.1. Chemical Reagents

Analytical grade chemicals were used during the whole experimentation. Distilled water was used during membrane casting and other processes. Polyethersulfone (58,000 M_w_) was acquired from Ultrasone, Germany. N, N-Dimethyl acetamide (DMAc) with a molecular weight of 87.12 was used as a solvent and obtained from Fisher Scientific, UK. The 75,000 MW chitosan was purchased from Sigma Aldrich. Polyvinylpyrrolidone (PVP, M_w_ 40,000 g/mol) was purchased from Merck, Germany. Activated carbon was purchased from Merck, Germany.

### 2.2. Methodology

The phase inversion method was used to synthesize the membranes as shown in Figure 1. Two batches of membranes were synthesized using 16% wt. and 20% wt. polyethersulfone (PES). PES was added in the DMAc solvent while continuously stirring in a media bottle to create a casting solution. The solution was stirred for 24 h to ensure homogenous mixing at room temperature. The casting solutions were composed of different combinations of PES, PVP as a pore former, chitosan, activated carbon and thiolated chitosan. The details for the casting solutions are given below in Table 1. A polypropylene support was fixed firmly onto a glass slide onto which these mixtures were then casted using a Filmograph elcometer, and were dipped into cold water (5 °C) immediately and were left into the water for 15 min so that the membrane casted completely. Membranes were then washed with distilled water and were wrapped in filter paper to dry overnight.

### 2.3. Characterization Techniques

Scanning Electron Microscopy was done using a JEOL-JSM-6490LA, which had a working distance of about 10 mm, operating voltage 10–20 kV and 35–60 spot size. The membranes were cut into 1 cm^2^ pieces and were placed into liquid nitrogen for drying and clean breaking into small pieces so that the cross section is not disturbed. ATR-FTIR was done to analyze the functional groups on the membrane. The dried membranes were cut into 0.5 × 0.5 cm^2^ dimensions for ATR-FTIR. The spectral range was between 500 and 3500 cm^−1^ and the resolution was about 2 cm^−1^. The practice was done on a BRUKER model: ALPHA II of the FTIR spectrophotometer. For optical profilometry, 0.25 × 0.25 cm^2^ pieces of membrane were cut and pasted onto a glass slide. The slide was then placed onto the stage and using the profilometer, the surface was scanned. A NANOVEA PS-50 optical profilometer was used for this purpose to measure the roughness of the membrane samples. The contact angle depicts a surface’s wettability. The most common method for contact angle measurement is the sessile drop method. The dried membrane sample of roughly 1 × 1 cm^2^ was vaccinated with a deionized water droplet. The angle between the membrane surface and the droplet was determined at three random points for minimal error. The equipment used for this purpose was a DSA-25 drop shape analyzer by KRUSS. Water retention measures the ability of the membrane to absorb water. This measurement was done by soaking 2 cm^2^ pieces of the membrane in distilled water for 24 h. The samples were then removed and excess water present on the surface was removed. The membranes were then weighed, and this weight was termed as the wet weight (W_h_). The membranes were then dried for 3 h at 40 °C under vacuum. The membranes were weighed again, and the weight was named as the dry weight (W_dry_). The values obtained were inserted into Equation (1) given below, to measure water retention (WR).
(1)$$\mathrm{WR}=\frac{W_{h}-W_{\mathrm{dry}}}{W_{w}}\times 100$$

For calculation of porosity, gravimetric analysis was done using Equation (2). Membranes were first dried at 50 °C for 3 h to remove any moisture and then weighed (W_d_). These dried membranes were then soaked into water for 24 h at room temperature. After 24 h, the membranes were removed from water and excess water droplets were wiped off carefully and weighed (W_w_). The values obtained were applied to the Equation (2). The method was repeated three times and the average values were calculated.
(2)$$\varepsilon =\frac{W_{w}-W_{d}}{{\mathrm{Al}}_{\rho }}\times 100$$
where A is the area of the membrane; ρ is the density of water, which is assumed to be 1 g/cm^3^; and (l) is the thickness of the membrane at 250 [25].

A tensile test was performed by using a Universal Testing Machine (UTS) (Shamizdu AG-X Plus). Membrane samples were cut into specified sizes (2.4 cm long and 1.4 cm wide). Flux rate was measured to evaluate the permeation flux of the membranes, which is the amount of fluid passing through a membrane under the influence of certain parameters, which include time, area, volume, etc. The membrane sample was place inside the filtration assembly that was attached to a vacuum and maintained at a 60 cmHg pressure. Distilled water was made to flow through the membrane and the time and volume was noted. The acquired values were then put into Equation (3).
(3)$$J=\frac{V}{\mathrm{AT}}$$
where J is the permeate flux calculated in Lm^−2^h^−1^, V is the water volume, A is the area of membrane and T is the time for permeation flux. For every membrane, three values were calculated for minimal error.

Bovine serum albumin (BSA) flux was calculated to gauge the anti-fouling properties shown by the membranes. A 1000 ppm solution of BSA was prepared by dissolving 1 g in 1 L water. This alkaline solution had a pH of 8.22. The ionic strength was calculated to be 0.1. This solution was projected through the synthesized membranes at a pressure of 0.1 MPa. Using a calibration curve, the concentration present in the permeate was calculated by measuring absorbance at 280 nm. This was done in agreement with Beer–Lambert’s law. A UV spectrophotometer was used for this purpose and the model was a SHIMADZU, UV. The BSA rejection was calculated by putting the required data into Equation (4).
(4)$$R(\%)=(1-\frac{C_{p}}{C_{f}})\times 100$$
where C_f_ and C_p_ are the initial and final concentrations (mg L^−1^) respectively.

## 3. Results

### 3.1. ATR-FTIR

The ATR FTIR spectra in Figure 2a–c for 16% wt. membranes, 20% wt. membranes and composite membranes, respectively, show most of the characteristic peaks of the PES polymer. In Figure 2a,b, it can be see that all membranes of both concentrations show peaks at 1580 cm^−1^ (strong C=C bending intensity shows benzene ring) and 1480 cm^−1^ (C-C bond stretching), which correspond to the polyethersulfone structure [26]. These peaks are also observed in Figure 2c for the composite membranes. Bands shown at 1200 cm^−1^ confirm C–O stretching of the ether and carboxylate structures. The aromatic ether group is identified by the C–O–C stretching at 1246 cm^−1^ in the polyethersulfone structure showing the presence of a sulfone group, which is confirmed by the literature [27]. The peaks at 1150 cm^−1^ could be attributed to the sulfonyl (O=S=O) group. At 873 cm^−1^, C–O and C–H vibrations indicate the presence of activated carbon. The absorbance peak at 1090 cm^−1^ corresponds to the –C–O–C– bonds, indicating the presence of chitosan. Two weak bands at 1322 and 1395 cm^−1^ are due to the methyl groups and are present exclusively in the spectrum of polysulfone [28,29,30,31]. As the concentration of polyethersulfone is a lot more than 1.25 wt % of the additives, for all the membranes in Figure 2a–c, it can be seen that there is no significant change observed in the spectra after addition of chitosan, activated carbon and thiolated chitosan.

### 3.2. Morphological Analysis

The micrographs shown in Figure 3 and Figure 4 show the cross sections of the membranes. These membranes are shown to have an asymmetrical assembly having a gradient-like structure. A dense top layer is observed in all the PES membranes. A porous sublayer consisting of finger-like structures that are also called channels are also observed. Better channel connectivity and more channel formation result in better permeability. Under this sublayer, a sponge-like mesoporous structure can also be found. With the addition of additives such as PVP and chitosan, a slight change in the morphology is observed. The addition of these fillers makes the solution more viscous as both PVP and chitosan have high molecular weights favoring lesser macrovoids formation [24]. With the introduction of activated carbon into the membranes, not only porosity increases but also the surface roughness. Usually with the addition of activated carbon, the pores widen/expand, which results in increased membrane porosity [27]. PVP, chitosan and thiolated chitosan are hydrophilic in nature [28]. The finger-like structures are more evenly distributed and less macrovoids are observed in membranes with these fillers. This change in membrane structure is attributed to the hydrophilicity of these fillers causing a swift non-solvent and solvent (DMAc) exchange throughout the phase inversion process. This rapid exchange gives birth to wider channels (finger-like structures).

In comparison, there is a difference in 16% wt. and 20% wt. membranes as can be seen in Figure 3 and Figure 4. Generally, the increase in polymer concentration diminishes macrovoids and makes better finger-like structures. So, the membranes with an increased polymer concentration (20% wt.) show better channel structures, as shown in Figure 4. There is a reduction in macrovoids and increase in the fingerlike cavities. The addition of fillers such as PVP and chitosan have resulted in a denser top layer as compared to the pristine PES membrane. In the case of a pristine PES membrane, the active layer is on top whereas there is also an uneven region below, depicting the polyethylene/polypropylene fabric that was used as a support. Membranes with composites in both the concentrations show better channel formation.

For 16% wt. composite membranes there are more voids present (Figure 3) as compared to 20% wt. composite membranes; the reason being the concentration of the polymer. As it increases, the casting solution becomes more viscous, resulting in formation of a membrane having lesser voids. The addition of thiolated chitosan also resulted in pronounced and smooth channel formation, as shown in Figure 4.

### 3.3. Surface Hydrophilicity

The wettability of a membrane shows its hydrophobic or hydrophilic nature, which is one of the most substantial pieces of information about a membrane. One of the major factors that govern the permeability rate for membranes is their wettability. The higher the wettability, the higher hydrophilic character, and the higher the enhanced flux rate and anti-fouling property [2]. This property is evaluated by the measurement of the contact angle. Whereas, surface energy can be defined as an intermolecular force present on the material surface; it is the determinant for the extent of forces either repulsive or attractive that are exerted onto the other surface. There exists an inverse relation between the contact angle and surface energy [32]. The graph plotted between these two values prove that with an increase in contact angle, there is a decrease in surface tension and vice versa. For a surface to be hydrophilic, the contact angle should be less than 90°. Figure 5 shows that pristine PES membranes have a higher contact angle, showing less wettability properties. With the addition of fillers such as PVP and chitosan for both concentrations, there was a drop in the contact angle values as both of these fillers are hydrophilic themselves. When these additives are added to the membrane solution, some part of them remains on the surface while casting. This presence on the surface of the membranes induces hydrophilic groups on the surface; hence, decreasing the contact angle and increasing the surface energy. For membranes incorporated with activated carbon, the increase in contact angle is more pronounced for both concentrations as shown in Figure 5a,b. This is the result of the presence of activated carbon on the membrane surface, as activated carbon is highly hydrophobic.

Some reports also claim that a decrease in contact angle results in better anti-fouling activity [5]. As shown in Figure 5c for the composite incorporated membranes, the contact angle of 52.2 for 16% wt. and 59.85 for 20% wt. shows that the addition of both chitosan and activated carbon has resulted in a balance of hydrophilic and hydrophobic functional groups on the membranes’ surfaces. The contact angle of the 20% wt. composite incorporated with an unmodified chitosan membrane is slightly more. A decrease in contact angle is seen for this membrane when unmodified chitosan is replaced with thiolated chitosan. This is due to the higher concentration of PES as PES itself is more inclined towards the hydrophobic spectrum. The concentration for thiolated chitosan was uniform at 1.25 wt. % but the concentration of polyethersulfone was increased from 16 wt. % to 20 wt. %, leading to a slight drop in the contact angle. The –SH group on the chitosan thiomer has better hydrophilic abilities than that of the chitosan. All in all, the combination of these additives has achieved the desired aim to reduce the hydrophobicity of the membranes to a great extent.

### 3.4. Surface Roughness

As shown in Figure 6, the highest surface roughness was observed for membranes incorporated with thiolated chitosan. This is the effect of the thiol group hydrophilicity. During phase inversion, increased hydrophilicity impacts the phase exchange process and accelerates it, resulting in immigration of the thiolated chitosan particles on the surface. Membranes incorporated with activated carbon showed increased surface roughness in Figure 6a. This is due to the presence of activated carbon particles on the surface. The porous activated carbon particles present on the surface are the reason for it [33]. The lowest roughness was for the 20% wt. composite membrane. This is due to the polymer concentration. With an increase in polymer concentration, less defects are formed. The high surface roughness for the PES/PVP/activated carbon membrane is due to the highly porous structure of the AC.

### 3.5. Water Uptake

As per Figure 7, in case of pristine PES, the percentage for water retention was lowest because of the hydrophobic nature of PES. The SEM micrographs also showed the asymmetric top layers for pristine PES membranes to be less dense than with the fillers, which is also a reason for the lower water uptake. The highest water retention ability was shown by the membrane with thiolated chitosan at 76.5%. The results contribute to the fact that with an increase in hydrophilicity, the water retention abilities of the membranes are also increased [34]. As per Figure 7b, all the composite membranes showed remarkably better performance than the pristine PES membranes. Addition of activated carbon had a positive impact. Even though it is hydrophobic, it has a highly porous structure. This increased porosity has played a role in better water retention [35].

### 3.6. Porosity

Porosity of a membrane is a very important factor. It plays its role in the permeation ability, adsorption abilities and antifouling properties of the membranes. The membranes should be porous enough for good permeation flux. Hydrophilic fillers can play an important role in the porosity of membranes [36]. According to Figure 8b, in composites, the highest porosity percentage was observed in the 16% wt. composite membrane at 85%, closely followed by 79% porosity of the 20% wt. thiolated chitosan-induced membranes. The 20% wt. membranes with chitosan also showed a good porosity percentage of 73% but was relatively low when compared with the other two composite membranes, as shown in Figure 8. This is a result of an increased polymer concentration, resulting in a more viscous casting solution, but the concentration of the pore former (PVP) was not increased. This leads to a reduced mean pore radius.

As per Figure 8a, membranes incorporating activated carbon also show a pronounced mean pore radius, with 73% porosity for both the concentrations owing to the highly porous structure of activated carbon. Another reason is the expansion of the pores in the membrane because of the activated carbon. These widened pores are good for the flux rate but in case of porosity, instead of an increase in the mean pore radius, the existing pores are widened with addition of activated carbon [30]. Chitosan membranes showed lower porosity because they possess a very high molecular weight and the casting solution becomes very viscous with even 1.25% addition of chitosan.

### 3.7. Tensile Strength

Membranes are employed under different operating conditions. Membranes with good mechanical strength are currently needed, where synthesis of multifunctional membranes is so prevalent. Membranes having a high polymer concentration possess lesser voids and defects in their structure, reducing the available sites for cracks to initiate or propagate. So, membranes with a greater concentration of polymer show better tensile strength, as can be seen in Figure 9. The highest value for UTS was shown by the 20% wt. composite membrane at 41.39 MPa. The lowest value was observed in the 20% wt. membrane, with thiolated chitosan at 15.11 MPa. The values for membranes with a PVP pore former were also low compared to pristine PES membranes. This is due to the solubility of PVP and thiolated chitosan, as both of these are soluble in water [14,37]. Using fillers that are soluble in water may result in reduced mechanical properties, as during immersion, these soluble molecules tend to increase the macrovoid formation, compromising the mechanical strength of the membranes. The materials with higher water solubility dissolve in water. This leads to the formation of voids where the material resided. These voids then provide the sites for the crack to propagate further, resulting in membrane tear.

Incorporation of activated carbon particles results in increased porosity of the membrane. Activated carbon has a highly porous structure, which also makes it an excellent adsorbent. This high mean pore radius of activated carbon can also lead to broadening of the pores. The activated carbon gets dispersed into the finger-like structures and forms smaller pores. Activated carbon can make agglomerates in the membrane structure, as reported by Hwang et al.; this agglomeration is also a reason for reduced mechanical strength [30]. Mechanical properties such as the elastic modulus also decreases with a sudden increase in porosity because of a high molecular weight filler as shown in Figure 10 [37]. In the case of elastic modulus, activated carbon is not a polymer and its elastic region is very low as compared to the polymers. This reduced elastic region is a reason for the decrease in elastic modulus.

### 3.8. Water Flux and Bovine Serum Albumin Flux

As can be seen in Figure 11 for the 16% wt. PES membranes, the lowest water flux was shown by membranes incorporating activated carbon. This is due to the hydrophobic nature of the activated carbon. Although it has a porous structure, the water-repelling ability of activated carbon hinders the flow of water through the membrane. With the induction of activated carbon, the surface of the membrane becomes more hydrophobic when compared to a pristine PES membrane, resulting in a decrease in water permeating through the membrane. Both the PES and activated carbon are hydrophobic in nature, and with only 1% wt. PVP incorporated into the membrane, there is no significant effect on the hydrophobicity of the membrane. The best permeability rate was shown by membranes incorporating chitosan and PVP, as both these fillers are highly hydrophilic, thus affecting the performance of the membrane in a positive way. The membrane with the composite (containing both the fillers, i.e., activated carbon and chitosan) showed a higher permeability rate than the pristine PES membranes and membranes having AC, but less than the chitosan-incorporated membranes. This is due to the presence of activated carbon in the membrane, as it lends its porosity and active sites to the membrane. The trend remains almost the same for the membranes with a 20% wt. PES concentration. There is a slight drop in the flux rate of the 20% wt. membranes as compared to the 16% wt. membranes. This drop is because of an increased amount of polyethersulfone, as it is closer to the hydrophobic end of the spectrum and is the reason for the lesser flux, comparatively [38,39].

The highest BSA flux was shown by membranes incorporating activated carbon as a filler. The composite activated membranes also showed a greater BSA flux rate, only second to the AC-incorporated membranes. This is due to the highly porous nature of the activated carbon. Due to its porosity, the pores are expanded, which is the reason that the protein could not clog the pores. However, the BSA flux was lower for membranes with PVP and chitosan as compared to the ones with activated carbon. This is the result of the hydrophilicity of the membrane. It is possible that some part of the albumin protein present in the feed solution was trapped in the pores and the flux rate was affected by it. The 20% wt. membranes incorporated with thiolated chitosan and activated carbon also showed a reasonably good BSA flux rate compared to the other composite membranes. As thiolated chitosan and PVP both are hydrophilic and activated carbon is hydrophobic, a balance seems to have been reached, resulting in a good permeation rate and BSA flux.

## 4. Conclusions

Polyethersulfone (PES) membranes were fabricated by incorporating additives and composites. The membranes were prepared under two concentrations, 16% wt. and 20% wt., by the phase inversion method. A novel thiolated chitosan/activated carbon composite was introduced into a 20% wt. polyether sulfone ultrafiltration membrane. ATR-FTIR results confirm the functional groups for PES. The membranes incorporating the thiolated chitosan/activated carbon (PPTCAC) showed an improved hydrophilic character compared to the pristine PES membrane (P_0_), which was considered to be a reference membrane. The contact angle for the 20% wt. PPTCAC was reduced to 55.7 from the 65 of the 20% wt. P_0_. The pure water flux increased from 105 L/m^2^h for the 20% wt. P_0_ to 114 L/m^2^h for the 20% wt. PPTCAC. A remarkable increase in antifouling properties was also observed as the BSA flux was calculated at 51 L/m^2^h for the 20% wt. PPTCAC, which was considerably higher when compared to the 20% wt. P_0_ at 23 L/m^2^h. The cross-section images of the Scanning Electron Microscopy exhibited enhanced finger-like structures for the 20% wt. PPTCAC, which helped give a better flux rate. Yield point decreased for the 20% wt. PPTCAC to 3.04 from the 4.09 of the 20% wt. P_0_, as it is water soluble and results in formation of macrovoids, leading to reduced mechanical strength. The water retention properties showed great improvement from 22.84% for the 20% wt. P_0_ to 76.5% for the 20% wt. PPTCAC. Surface roughness saw an increase of 20% wt. for PPTCAC at 2690 nm, whereas the surface roughness observed for the reference membrane was 832 nm. The composite-incorporated membrane displayed a substantial increase in mean porosity, which was measured using gravimetric analysis. The porosity increased to 79% for the 20% wt. PPTCAC from the 31.7% of the reference membrane. These results demonstrate that the thiolated chitosan/activated carbon composite shows great promise in the field of membrane technology and has potential as an antifoulant, and must be explored further.

## Figures and Tables

**Figure 1 membranes-11-00827-f001:**
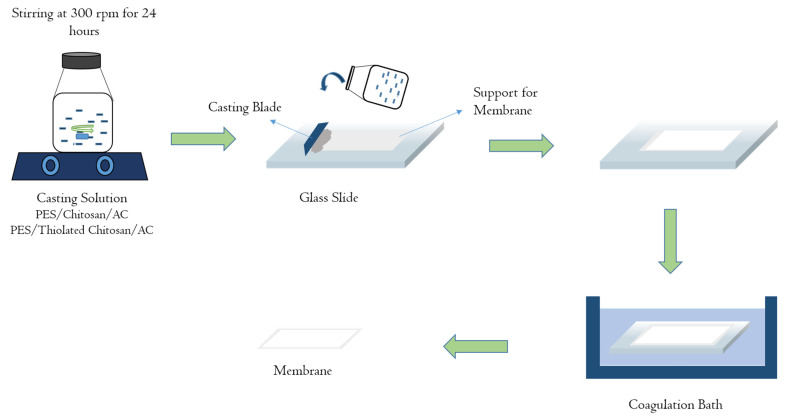
Schematic for the membrane fabrication process (phase inversion).

**Figure 2 membranes-11-00827-f002:**
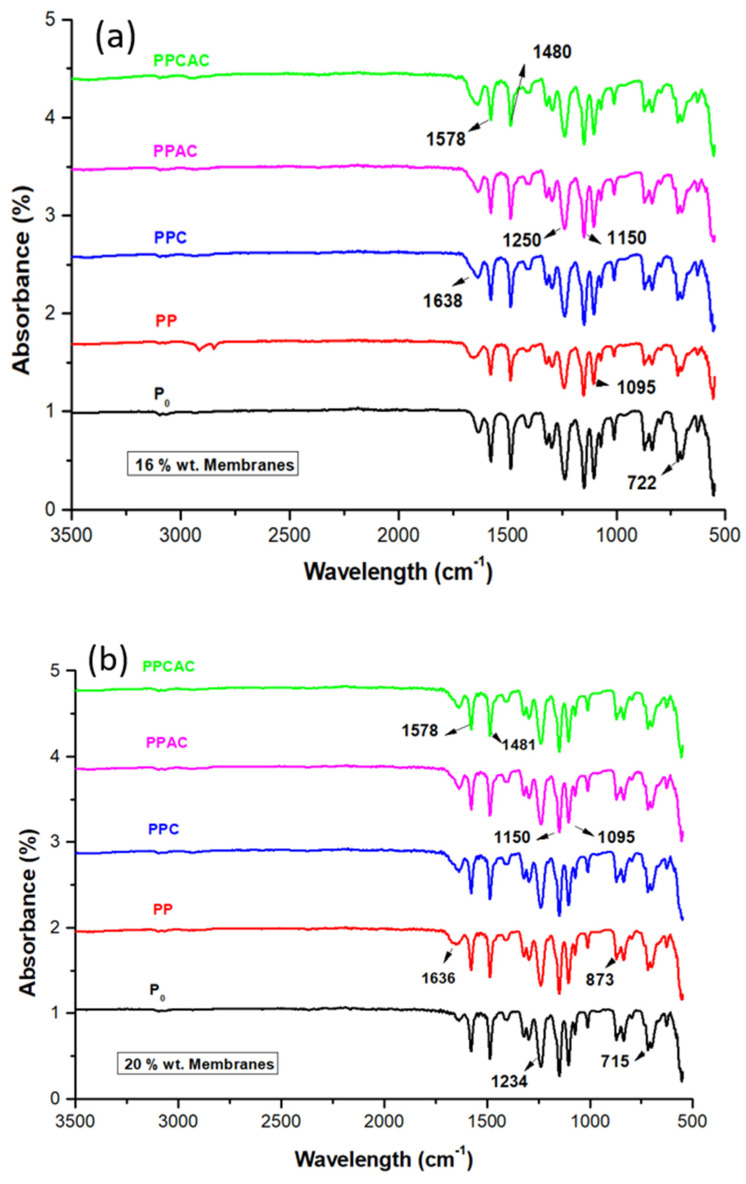
ATR-FTIR spectra for (**a**) 16% wt. PES membranes, (**b**) 20% wt. PES membranes and (**c**) composite membranes.

**Figure 3 membranes-11-00827-f003:**
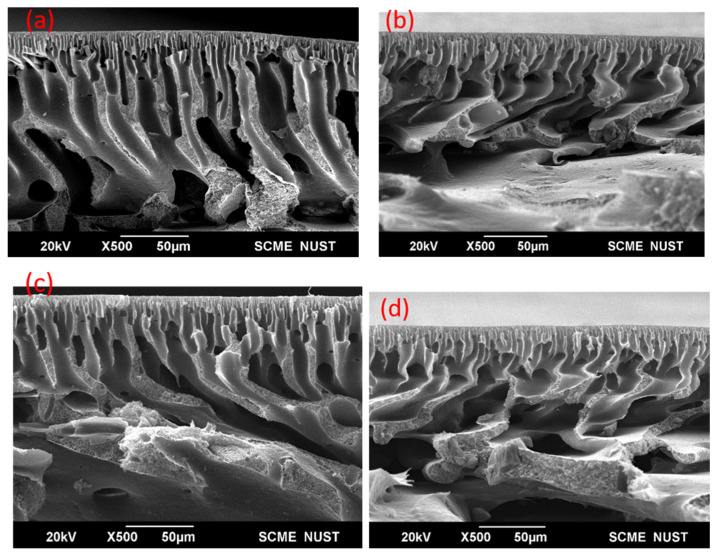
Cross-sectional SEM micrographs for the 16% wt. membranes where (**a**) P_0_, (**b**) PP, (**c**) PPC, (**d**) PPAC.

**Figure 4 membranes-11-00827-f004:**
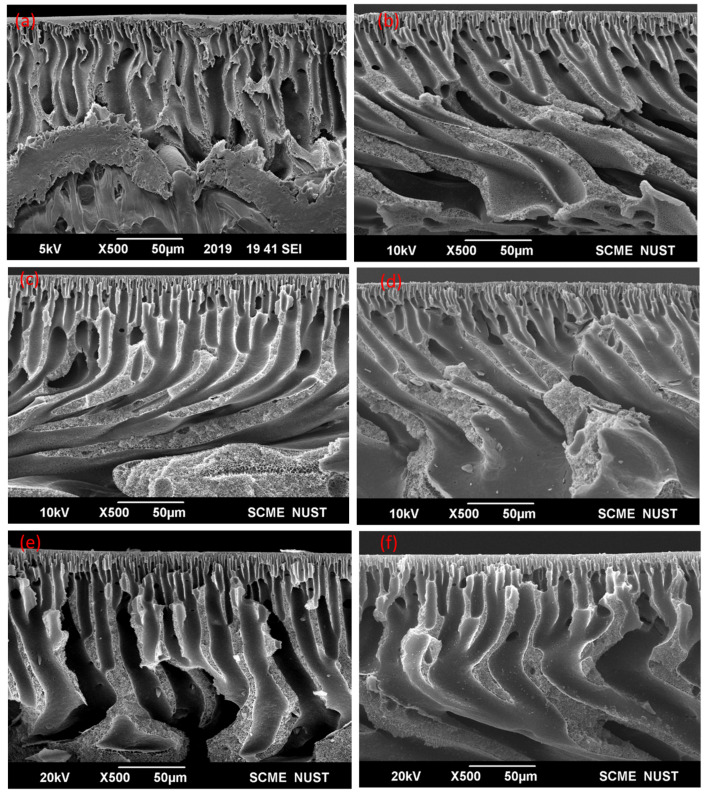
Cross-sectional SEM micrographs for 20% wt. membranes where (**a**) P_0_, (**b**) PP, (**c**) PPC, (**d**) PPAC, (**e**) PPCAC and (**f**) PPTCAC.

**Figure 5 membranes-11-00827-f005:**
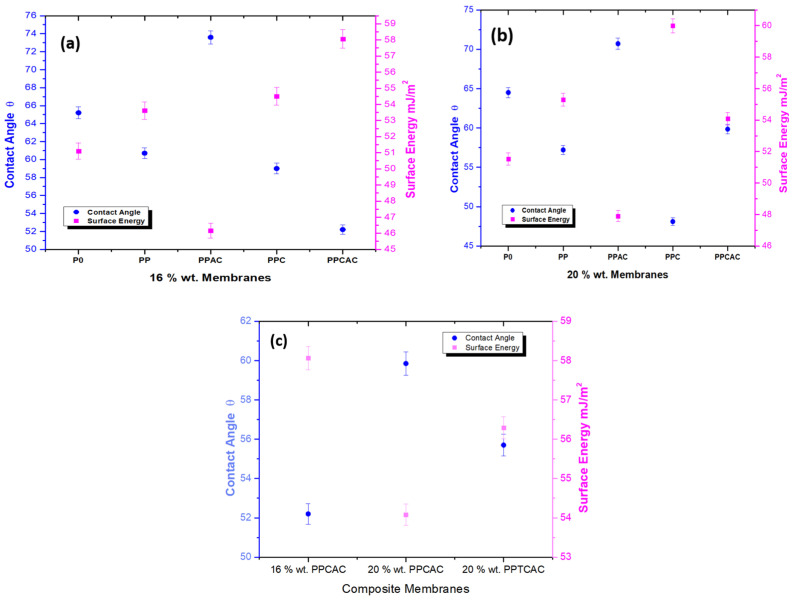
Contact angle vs. surface energy for (**a**) 16% wt. membranes, (**b**) 20% wt. membranes and (**c**) composite membranes.

**Figure 6 membranes-11-00827-f006:**
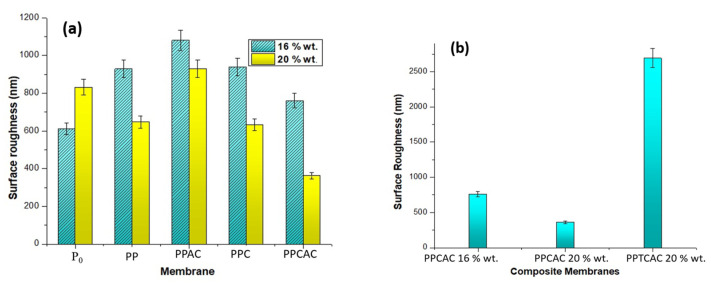
Surface roughness bar graph for (**a**) 16 and 20% PES membranes, and (**b**) composite membranes.

**Figure 7 membranes-11-00827-f007:**
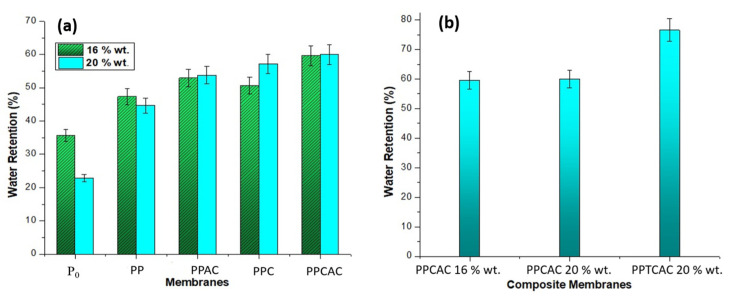
Water retention bar graph for (**a**) 16 and 20% PES membranes, and (**b**) composite membranes.

**Figure 8 membranes-11-00827-f008:**
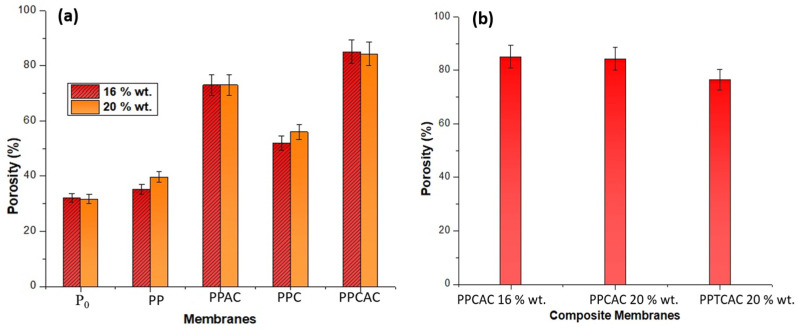
Porosity bar graph for (**a**) 16 and 20% PES membranes, and (**b**) composite membranes.

**Figure 9 membranes-11-00827-f009:**
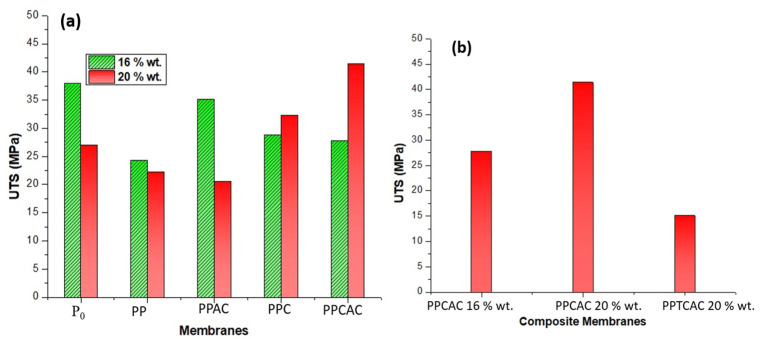
Ultimate Tensile Strength for (**a**) 16 and 20% PES membranes, and (**b**) composite membranes.

**Figure 10 membranes-11-00827-f010:**
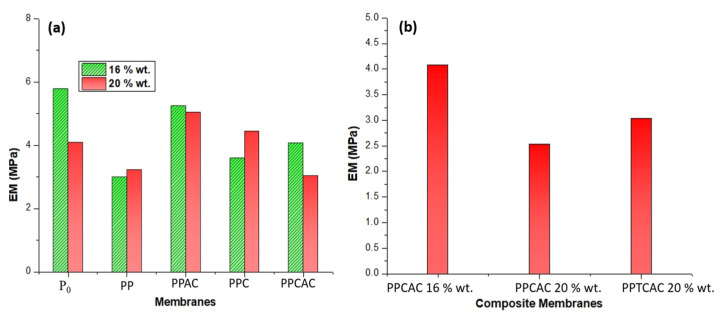
Elastic modulus graph for (**a**) 16 and 20% PES membranes, and (**b**) composite membranes.

**Figure 11 membranes-11-00827-f011:**
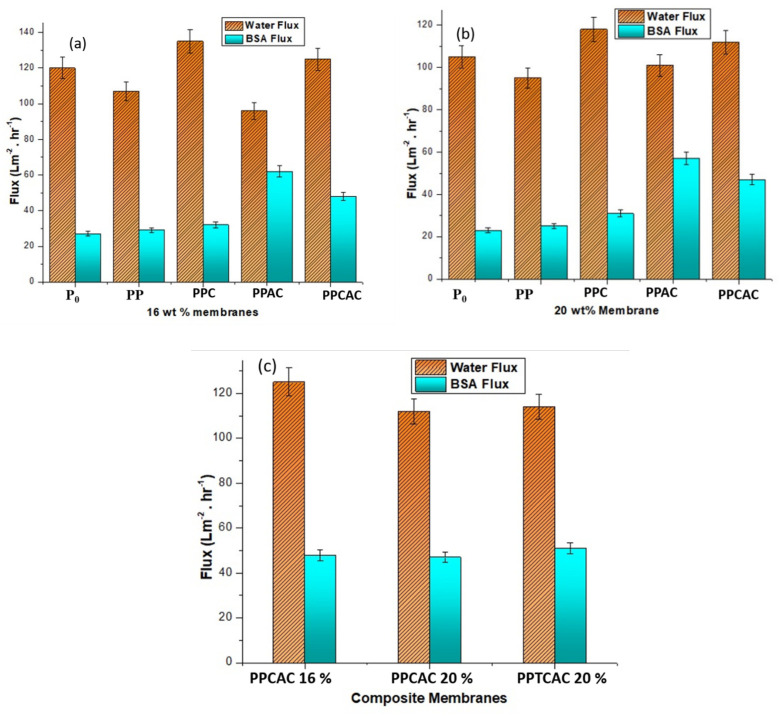
Water flux and BSA flux results for (**a**) 16% wt. membranes, (**b**) 20% wt. membranes and (**c**) composite membranes.

**Table 1 membranes-11-00827-t001:** Composition of the polyethersulfone membranes.

Sample	16% wt. PES			20% wt. PES			
	PVP (%)	Chitosan (%)	Activated Carbon (%)	PVP (%)	Chitosan (%)	Activated Carbon (%)	Thiolated chitosan (%)
P_0_	-	-	-	-	-	-	-
PP	1	-	-	1	-	-	-
PPC	1	1.25	-	1	1.25	-	-
PPAC	1	-	1.25	1	-	1.25	-
PPCAC	1	1.25	1.25	1	1.25	1.25	--
PPTCAC	-	-	-	1	-	1.25	1.25

## Data Availability

All the data is provided in the manuscript and will be available for the readers.

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
