# Peer review of "Anti-Foulant Ultrafiltration Polymer Composite Membranes Incorporated with Composite Activated Carbon/Chitosan and Activated Carbon/Thiolated Chitosan with Enhanced Hydrophilicity"

_membranes, 2021, doi:10.3390/membranes11110827_

Round 1
Reviewer 1 Report
In my opinion, in this work authors propose a modification for ultrafiltration membranes, with a basic characterization. With the shown results it is clear the structural changes, but I think functional characteristics are not enough demostrated.
I would like to ask some questions:
1.- I could not find any information about chitosan. It is commercial? In this case, why it is not included in 2.1 section?
2.- Line 164: Wdry is not use in WR equation. Is it the same that Wd of that equation? The protocol for Wd is different for “Water retention” than for “Porosity” but it is the same symbol. Is it the same value?
3.- The protocol for wet membrane is the same for “Water retention” than for “Porosity”, however you use different variables. Why? Which is the different between Ww and Wh?
4.- Line 176: The variables use in the denominator of equation of porosity are not explained. I suppose that correspond to Area*thickness*density but it must be detailed
5.- It would be nice a scheme of the dispositive to measure the flux (Flux rate and BSA flux). The results obtained are for initial time? After some minutes?
6.- The BSA flux measurements are not representative of fouling behaviour. Dis you study the flux decrease in the time? The BSA flux data at which time correspond?
7.- For roughness surfaces (as PPC) it is difficult to get contact angle with small dispersion. The bar error of Figure 6 are right? Furthermore, for the hydrophilic surface the time between the drop deposit and the contact angle is measured is important. Is the protocol always the same for all samples? Can you explain it?
8.- In my opinion table 3 and 4 are redundant. All data have been shown in previous figures.
Author Response
Dear reviewer,
We have tried to address your comments in most appropriate way. We hope that the answers will be satisfactory for you.
Regards

Reviewer 2 Report
Manuscript entitled “Anti-foulant Ultrafiltration Polymer Composite Membranes Incorporated with Composite Activated Carbon/Chitosan and Activated Carbon/Thiolated Chitosan with Enhanced Hydrophilicity” submitted by Syeda Samia Nayab, M. Asad Abbas, Shehla Mushtaq, Bilal Khan Niazi, Mehwish Batool, Gul Shehnaz, Naveed Ahmad and Nasir M. Ahmad, can be accepted for publication in Membranes Journal, after a minor revision.
Here is a list of my specific comments:
- Page 1, Abstract: This section is quite too general. Include here the most important experimental results to highlight the importance of this study.
- Page 5, line 210: “The C-O-C stretching was…”. This paragraph should be detailed.
- Page 7, 3.2. Morphological Analysis: This section should be systematized and the structural particularities should be clearly explained.
- Line 298: “A decrease in contact angle can…”. This variation should be clearly explained.
- Line 378: “Incorporation of activated carbon particles…”. This observation should be detailed.
Author Response

(The authors gave the same response as above.)

Reviewer 3 Report
In this work, the authors synthesized the ultrafiltration polyethersulfone membranes by incorporating chitosan, activated carbon, and thiolated chitosan. The resulting membranes were analyzed via using a series of characterizations, and some results were interesting. However, there are some issues that should be addressed. I suggest a major revision. Details are below:
- Please do not use the abbreviations in Abstract, such as ATR-FTIR, SEM, BSA (Bovine Serum Albumin).
- In Abstract, “SEM micrographs also showed better channels for permeability with the addition of composites reaffirmed by flux rate.” It cannot be obtained these results only by the SEM images.
- Line 79. What’s the “Na2+”?
- Please check “Polyethersulfone (58000 Mw)”, “Polyvinylpyrrolidone (PVP, Mw 40,000 g/mol)”.
- The schematic for the membrane fabrication process in Figure 1 is not clear.
- Please check the equation of water retention WR.
- In Figure 4, there are two (a) and two (b). Please revise them.
- In Figure 6. Why the authors used the fitting curve in the figures?
Author Response

(The authors gave the same response as above.)

Round 2
Reviewer 1 Report
I agree with the author's comments. Although only some of them are included in the new version of the paper. It could be interesting include also all of them